# Concordance between self-report and six commonly used clinical estimates or serological measures: Insights from a Canadian healthy aging study

Tovan Lew[1,2], Tetiana Povshedna[1,2,3], Elizabeth M. King[4,5,6,7], Shelly Tognazzini[6], Angela Kaida[4,6], Melanie C. M. Murray[2,4,5,7,8], Hélène C. F. Côté[1,2,3,4,8*], on behalf of the British Columbia CARMA-CHIWOS Collaboration (BCC3; CIHR CTN 335)[¶]

1 Department of Pathology and Laboratory Medicine, University of British Columbia, Vancouver, British Columbia, Canada, 2 Edwin S.H. Leong Healthy Aging Program, University of British Columbia, Vancouver, British Columbia, Canada, 3 Centre for Blood Research, University of British Columbia, Vancouver, British Columbia, Canada, 4 Women's Health Research Institute, Vancouver, British Columbia, Canada, 5 Oak Tree Clinic, British Columbia Women's Hospital and Health Centre, Vancouver, British Columbia, Canada, 6 Faculty of Health Sciences, Simon Fraser University, Burnaby, British Columbia, Canada, 7 Department of Medicine, Division of Infectious Diseases, University of British Columbia, Vancouver, British Columbia, Canada, 8 Experimental Medicine, University of British Columbia, Vancouver, British Columbia, Canada

¶ The complete membership of the author group can be found in the Acknowledgments.
* hcote@pathology.ubc.ca

## Abstract

### Objective

To assess the concordance between self-reported and clinically assessed prevalence of selected chronic conditions and latent viral infections among women living with and without HIV.

### Methods

Women (aged ≥ 16 years residing in British Columbia) enrolled in the BCC3 Study, a prospective cohort, between 2020 and 2024 were included in the cross-sectional analysis. Self-reported prevalence of six conditions/viruses (chronic kidney disease, liver disease, depression, post-traumatic stress disorder, and hepatitis B and C viruses (HBV, HCV)), were compared to clinical estimates based on screening tools and serology. Agreement was assessed via Cohen's kappa.

### Results

In both women with (n = 220) and without HIV (n = 309), clinical estimate-based prevalence of depression and PTSD was higher than self-reported prevalence (all p < 0.001). Among women with HIV, clinical estimate-based prevalence of HBV was higher than self-report-based prevalence (p < 0.001). For both groups, there was no

**Data availability statement:** The data for this study contain potentially identifying and sensitive patient information. Given that a substantial proportion of the dataset is derived from individuals living with HIV, a highly stigmatized condition, strict confidentiality must be maintained. In addition, the dataset includes data from Indigenous participants, for whom principles of community consultation and governance apply prior to any public data sharing, even in deidentified form. Data are available upon request from the corresponding author, or the British Columbia Women's Hospital via email (bcc3@cw.bc.ca), for researchers who meet the criteria for access to confidential data.

**Funding:** The British Columbia CARMA-CHIWOS Collaboration (BCC3) has received funding support from the Canadian Institutes of Health Research (CIHR) project grants (PJT-162348, PJT-175006), Community-based research grant (CBR-170103), and Women's Health and Mentorship grant (F19-05017); CIHR HIV Clinical Trial Network support (CTN 335) and pilot grant funding (CTNPT046, CTNPT050); UBC Partner Recognition Fund; UBC Community University Engagement Support Fund, and UBC Public Scholar Initiative; Simon Fraser University's Community Engagement Initiative; and Michael Smith Foundation of Health Research (trainee award). AK received a salary support award from the Canada Research Chair Program. MCMM received salary support from Michael Smith Foundation of Health Research Health Professional investigator award. TP received funding from the UBC Four Year Doctoral Fellowship, UBC Centre for Blood Research, and Mitacs Globalink Graduate Fellowship. TAGL received funding from the UBC Edwin S.H. Leong Centre for Healthy Aging Summer Student Research Award. The funders had no role in the study design, data collection and analysis, decision to publish, or preparation of the manuscript.

**Competing interests:** The authors have declared that no competing interests exist.

difference between the two prevalence estimates for chronic kidney disease. Among women without HIV, clinical estimate-based prevalence of liver disease was lower than self-report-based prevalence ($p < 0.001$), but this was not the case for women with HIV. In both groups, agreement between self-report and clinical estimate of prevalence was fair to poor for all conditions/viruses (all $\kappa < 0.4$), except for HCV, for which the agreement was near perfect ($\kappa > 0.8$).

## Conclusions

Self-reported HCV history shows high concordance with serology, but the same is not true for HBV. The prevalence of liver disease, kidney disease, depression, and post-traumatic stress disorder as reported by participants may differ from clinical estimates. Our findings highlight the complexity of aligning self-report data with clinical estimates and suggest that both types of data should be used for a comprehensive assessment of prevalence in research.

## Introduction

Although international initiatives aim to provide equal care and disease outcomes for all people living with HIV, women, who comprise over half of the global population living with HIV [1], continue to face worse health outcomes compared to men, particularly in North America [2]. Women living with HIV experience a greater burden of age-related comorbidities (hereafter concurrent conditions) and are more likely to experience physical and mental concurrent conditions or both than men living with HIV and women without HIV [3–6]. A recent study also reported a higher prevalence of non-HIV chronic/latent viral infections among females compared to males with HIV [7], a factor that could modulate concurrent conditions. Sex and gender differences play a crucial role in the wellbeing of women living with HIV [8], who experience significant socio-structural and psychological stressors including discrimination, gender-based violence, poverty, and isolation, compared to men [9]. Despite improvements in treatment, women living with HIV in Canada have a life expectancy 5–10 years shorter than women without HIV and 7 years shorter than men living with HIV [10]. As life expectancy between men and women living with HIV has widened in recent decades in British Columbia [11], accurately characterizing the prevalence of health conditions in this population is crucial to promoting healthy aging and improving overall aging experiences in women.

Cohort studies rely heavily on self-report data regarding common concurrent conditions, including chronic kidney disease (CKD), liver disease, depression, post-traumatic stress disorder (PTSD), heart and lung disease, and chronic/latent viral infections [7]. However, self-report data in the general population can be inconsistent with registered medical data and clinical estimates derived from screening tools and/or serology, producing over- or under-representation, depending on the condition and cohort studied [12,13]. Prior studies using Cohen's kappa and related agreement metrics have demonstrated that agreement tends to be higher

for serologically confirmed infections and lower for mental health and chronic conditions, particularly in marginalized populations [12,14,15]. In cohorts of people living with HIV, previous work has reported fair-to-moderate agreement for several comorbid conditions when comparing self-report with clinical or administrative data [16,17]. Such discordance can compromise the accuracy of prevalence estimates, which can in turn influence the effectiveness of health-promoting interventions and research findings informed by self-reported data. In the context of women living with HIV, discrepancies in self-reported data can signal opportunities for health literacy promotion and improved provider-patient communication. Ensuring that women are aware of their diagnoses helps them make informed decisions and can positively affect treatment adherence and ultimately quality of life. However, few studies have systematically examined agreement across multiple physical and mental health conditions among women living with and without HIV using a consistent analytic framework.

Thus, the objective of this study is to assess agreement between self-reported and clinical estimates or serological measures of chronic kidney disease, liver disease, depression, post-traumatic stress disorder, hepatitis C virus, and hepatitis B virus among women living with and without HIV enrolled in the British Columbia CARMA–CHIWOS (BCC3) Study, using Cohen's kappa (κ), to inform ours and other HIV studies. As simple percent agreement does not account for chance concordance, Cohen's κ is used to assess chance-corrected agreement for binary categorical variables. Although κ can be influenced by condition prevalence, it remains a standard metric for evaluating concordance between self-reported and clinically derived health measures when interpreted in context [18,19].

## Methods

### Study and participants

This study employs a cross-sectional analysis conducted within the British Columbia CARMA–CHIWOS (BCC3) prospective cohort. BCC3 is a community-based, women-centered (cis- and trans-inclusive), holistic study investigating intersectional biochemical, clinical, and socio-structural factors as they relate to aging in women living with HIV [20]. Recruitment for the BCC3 study began in December 2020 and includes women living with HIV and socio-demographically similar women without HIV ≥ 16 years old in British Columbia, Canada. Eligible participants must be able to communicate in English, provide written informed consent, are not pregnant or breastfeeding, and are able to attend a clinical visit, usually at the BC Women's Hospital in Vancouver, or during recruitment drives in Prince George, Victoria, and Surrey. Participants who complete the clinical visit that involves clinical survey and biospecimen collection, then engage in a follow-up community visit facilitated by Community Research Associates with living experience of HIV and community expertise who are trained in research. The complete BCC3 questionnaire is available online at https://hivhearme.ca/ [20]. The study is approved by the harmonized research ethics board at the University of British Columbia (H19-00896).

Data is extracted for all participants enrolled as of July 2nd, 2024; those who did not complete either one of the surveys are excluded.

### Conditions of interest

The present study compares self-reported and clinical estimate-based prevalence of four concurrent conditions and two chronic/latent viral coinfections of interest. Our self-report questions inquire about conditions diagnosed by a care provider rather than a perception of having a condition. These include CKD, liver disease, depression, PTSD, hepatitis C virus (HCV), and hepatitis B virus (HBV). Self-reported conditions are determined through survey responses, whereas clinical estimates are derived using a combination of survey data and validated assessment/screening tools: the Kidney Failure Risk Equation [21] and albuminuria levels [22] for CKD, the FIB-4 index [23–25], for liver disease, the 10-item Center for Epidemiologic Studies Depression Scale (CES-D-10) [26] for depression, and the 6-item PTSD Checklist-civilian version (PLC-C) [27] for PTSD, applying established cutoff values. Prevalence of chronic/latent viral infections is assessed using serological testing at the BC Centre for Disease Control. These included anti-HCV and anti-HBVc total (antibody against

the hepatitis B core antigen). Venous blood samples collected from participants during the clinical study visit allow for this testing. A detailed breakdown of assessment questions and definitions can be found in Fig 1.

Socio-demographic characteristics were also self-reported: age, ethnicity, employment status, individual income, history of homelessness, substance use (opioids, non-opioid sedating drugs, stimulants, psychedelics), and tobacco smoking.

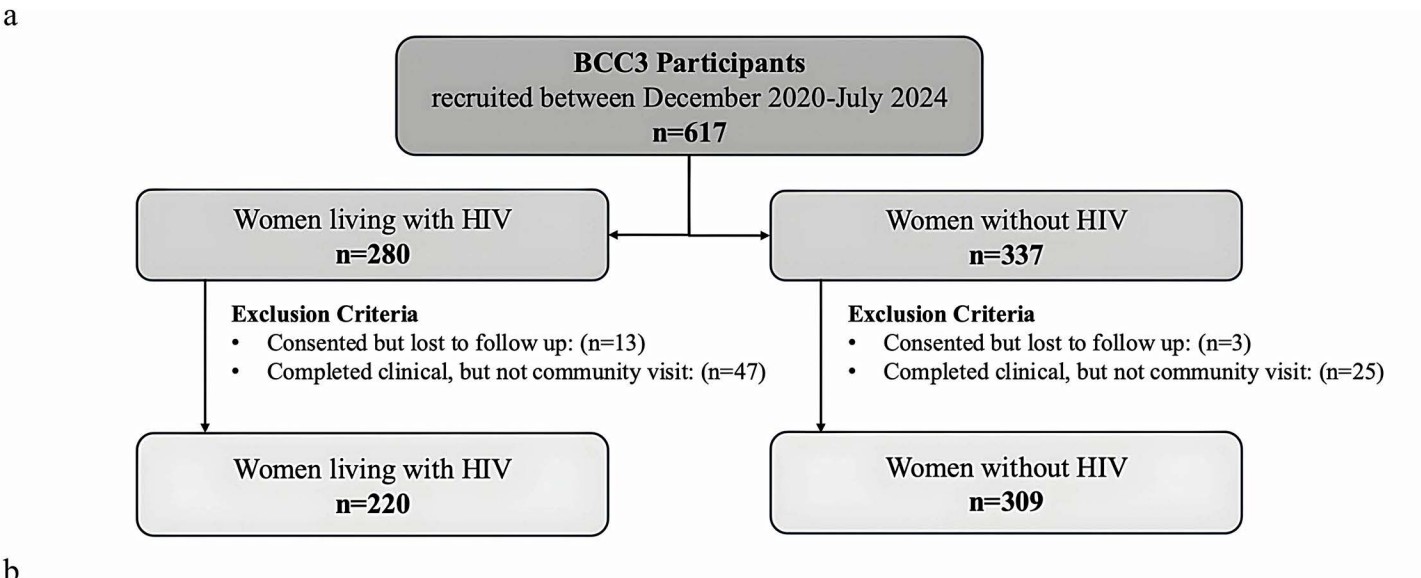

**Fig 1.** (a) Participant selection flowchart. (b) Self-report and clinical estimate assessment parameters for study condition/hepatitis viral infections. BCC3: British Columbia CARMA-CHIWOS Collaboration; PTSD: post-traumatic stress disorder; PLC-C: PTSD Checklist-civilian version; CES-D-10: Center for Epidemiologic Studies Depression Scale; KFRE: Kidney Failure Risk Equation; eGFR: estimated glomerular filtration rate; FIB-4: Fibrosis-4; HCV: hepatitis C virus; HBV: hepatitis B virus.

## Statistical analysis

Differences in sociodemographic characteristics and frequencies of categorical variables are assessed by unpaired t- or Mann-Whitney U, Chi-square, or Fisher's exact test, as appropriate, with Bonferroni correction (significance threshold p < 0.0031). Agreement is assessed using unweighted Cohen's kappa (κ), as agreement is evaluated between two binary categorical measures (presence vs absence) and a recognized heuristic interpretive scheme [28]. Cohen's kappa and corresponding 95% confidence intervals are calculated in GraphPad Prism (version 10.2.3) using equations 18.16 to 18.20 from Fleiss, Statistical Methods for Rates & Proportions [29].

## Results

### Demographic characteristics

Among 617 women enrolled between December 2020 and July 2024, 529 women (220 with and 309 without HIV) met inclusion criteria for the analysis (Fig 1); demographic characteristics are described in detail in S1 Appendix. Briefly, women were older within the HIV group with a median [IQR] age of 49 [41–58] vs 46 [34–57] years (p = 0.008), and the ethnicity makeup of both groups was significantly different (p < 0.0001). Compared to women without HIV, women living with HIV were more likely to report no current employment (55% vs 42%, p = 0.001), earnings under $20k/year (55% vs 44%, p = 0.019), and a history of homelessness (47% vs 37%, p = 0.02) although these were high in both groups relative to the general population. Tobacco use was prevalent in both groups, with 41% and 31% of women with and without HIV smoking, respectively, p = 0.003. Women with and without HIV reported similar substance use patterns, namely 20% and 17% current use, and 26% and 21% past use, respectively (p = 0.22). Missing data are described in S2 Appendix (≤3% for all variables).

### Agreement between self-report and clinical estimates

For CKD and liver disease, there was slight agreement between self-report and clinical estimate-based prevalences in both women living with and women without HIV, with Cohen's κ ranging from 0.05 to 0.20 (**Table 1**). In both groups, depression and HBV exhibited fair agreement, with κ ranging from 0.22 to 0.37. For PTSD, the agreement was slight in the HIV group (κ = 0.2) and fair in women without HIV (κ = 0.38). There was an almost perfect agreement for HCV in both groups (κ = 0.85 and κ = 0.91).

As skewed prevalence and marginal asymmetry can attenuate Cohen's κ despite high observed agreement, percent agreement and condition-specific positive and negative agreement are used to contextualize agreement patterns where appropriate. For chronic kidney disease, agreement was largely driven by concordant negative classifications, reflecting low condition prevalence rather than strong concordance in positive identification between self-report and clinical estimate. For liver disease, high overall agreement primarily reflected negative agreement, with relatively fewer concordant positive cases contributing to attenuation of Cohen's κ. For hepatitis B, skewed and higher clinical estimate based prevalence attenuated Cohen's κ despite high negative agreement, indicating systematic differences between data sources impacting concordance.

### Self-report versus clinical estimate based prevalence

Clinical estimate-based prevalence of depression and PTSD were higher than self-report-based prevalence in both groups (all p < 0.001). The same was true for the clinical estimate-based prevalence of HBV in women living with HIV (p < 0.001), but not in women without HIV (p = 0.097)

Clinical estimate-based prevalence of liver disease was lower than that based on self-report in women without HIV (p < 0.001), while there was no difference between the two in the HIV group (p = 0.68). Finally, for both groups, there was no difference between the two prevalence estimates for CKD and HCV.

**Table 1. Comparison and concordance of self-reported (SR) data and clinical based estimates (CE), among women living with HIV (n = 220) and women without HIV (n = 309).**

| Condition/Virus | Concordant (SR+CE +) n (%) | Concordant (SR – CE –) n (%) | Discordant (SR – CE +) n (%) | Discordant (SR+CE –) n (%) | SR-based prevalence n (%) | CE-based prevalence n (%) | Percent Agreement (%) | p-value | Cohen's Kappa (κ), (95% CI) |
|---|---|---|---|---|---|---|---|---|---|
| CKD WLWH, n (%) | 13 (6) | 138 (67) | 36 (17) | 21 (10) | 35 (16) | 49 (24) | 72.6 | 0.047 | 0.15 (<0.01-0.3) |
| CKD Women without HIV, n (%) | 6 (2) | 233 (77) | 33 (11) | 29 (10) | 35 (11) | 39 (13) | 79.4 | 0.6 | 0.05, (−0.08-0.17) |
| Liver Disease WLWH, n (%) | 7 (3) | 162 (80) | 15 (7) | 20 (10) | 27 (13) | 23 (11) | 82.8 | 0.684 | 0.19, (0.01-0.37) |
| Liver Disease Women without HIV, n (%) | 2 (1) | 264 (89) | 5 (2) | 25 (8) | 28 (9) | 7 (2) | 89.9 | **0.0003** | 0.08, (−0.06-0.23) |
| Depression WLWH, n (%) | 63 (30) | 76 (37) | 55 (27) | 12 (6) | 79 (36) | 120 (58) | 67.5 | **<0.0001** | 0.37, (0.26-0.49) |
| Depression Women without HIV, n (%) | 92 (32) | 105 (36) | 64 (22) | 28 (10) | 125 (41) | 163 (55) | 68.2 | **0.0005** | 0.37, (0.27-0.48) |
| PTSD WLWH, n (%) | 30 (14) | 101 (48) | 69 (32) | 13 (6) | 44 (20) | 101 (47) | 61.5 | **<0.0001** | 0.20, (0.08-0.31) |
| PTSD Women without HIV, n (%) | 66 (22) | 142 (47) | 80 (27) | 11 (4) | 80 (26) | 149 (49) | 69.6 | **<0.0001** | 0.38, (0.29-0.48) |
| HCV WLWH, n (%) | 74 (35) | 125 (58) | 7 (3) | 8 (4) | 85 (39) | 81 (37) | 93.0 | 0.693 | 0.85, (0.78-0.92) |
| HCV Women without HIV, n (%) | 40 (13) | 261 (85) | 3 (1) | 4 (1) | 44 (14) | 43 (14) | 97.7 | 0.9 | 0.91, (0.84-0.98) |
| HBV WLWH, n (%) | 14 (7) | 157 (74) | 34 (16) | 6 (3) | 21 (10) | 50 (23) | 81.0 | **0.0002** | 0.32, (0.17, 0.47) |
| HBV Women without HIV, n (%) | 3 (1) | 287 (93) | 13 (4) | 5 (2) | 8 (3) | 16 (5) | 94.2 | 0.097 | 0.22, (−0.01-0.46) |

SR: self-report; CE: clinical estimate; WLWH: women living with HIV; CKD: chronic kidney disease; PTSD: post-traumatic stress disorder; HCV: hepatitis C virus; HBV: hepatitis B virus; Fisher's exact test used for "*Liver Disease Women Without HIV*", all other *p*-values assessed using Chi-square tests. The Landis and Koch scheme is used for interpretation [28]. Missing data are described in S2 Appendix (≤3% for all variables).

## Discussion

### Principal findings

Our study found that concordance between self-report and clinical estimates among women living with and without HIV was condition and virus specific. Self-reported lifetime HCV history showed high agreement with serologic measures, whereas agreement for other physical and mental health conditions was lower, with similar patterns observed regardless of HIV status.

### Concordance for mental health and chronic physical conditions

Participants reported lower rates of depression and PTSD compared to clinical estimates, a finding that aligns with other studies that observed underreporting of PTSD in women with HIV [3], and fair (0.4) Cohen's Kappa for depression among women in the general population compared to data derived from medical charts [15]. Lower agreement for depression and PTSD may reflect differences between symptom-based screening tools and participants' understanding or recall of formal diagnoses, as well as stigma surrounding mental health conditions [30,31]. Untreated depression and PTSD can negatively affect the aging of women living with HIV in many ways, including by reducing adherence to antiretroviral therapy [32], stressing the importance of concordance between self-report and clinical estimates. While both prevalence

estimates of liver disease were similar among women with HIV, participants without HIV self-reported greater rates of liver disease compared to clinical estimates. The direction of this discrepancy elicits fewer concerns as it may reflect successfully treated liver conditions. As in previous studies of older adults in the general population [33], both groups self-reported comparable rates of CKD compared to clinical estimates, with a fair agreement between prevalences.

## Concordance for chronic viral infections

Our analysis also demonstrated subpar agreement between self-reported prevalence of two chronic/latent viral co-infections and corresponding serological results. Women living with HIV self-reported a lower prevalence of HBV relative to serology results indicating a history of natural HBV infection, a concern given higher likelihood of HBV reactivation in women living with HIV [34]. Discordance may arise from limited awareness of prior exposure detected through serology, particularly when infection was asymptomatic or resolved without clinical diagnosis [35,36]. In contrast, we observed near-perfect agreement for lifetime HCV prevalence in both groups, a reassuring finding demonstrating effective screening, diagnosis and communication between women and their care providers. However, willingness to self-report HCV and other infections may also depend on broader contextual factors, such as stigma around injection drug use, discrimination in healthcare settings, and the legal landscape regarding drug-related offenses and communicable disease disclosure. This study took place in British Columbia, Canada, and in settings where policies or social stigma are more pronounced, self-reported data on HCV and other conditions may underestimate true prevalence, reflecting barriers to disclosure rather than an absence of screening or knowledge about diagnosis [37].

## Contextual factors affecting concordance

Although subgroup analyses were not conducted, the socio-demographic context of this cohort is important for interpreting concordance patterns. High levels of socioeconomic disadvantage, substance use, and histories of homelessness in both groups may influence health literacy, access to care, and recall of diagnoses, all of which can affect the accuracy of self-reported data [12,14]. Prior studies have similarly shown that concordance between self-report and clinical data varies by demographic and social factors, particularly among women and marginalized populations [3,12,15]. While our findings may not generalize to populations with different socio-structural contexts, they underscore the importance of interpreting self-report data within the lived realities of the populations studied.

This report is strengthened by the availability of an established, well-characterized community-based cohort of women living with HIV, and socio-demographically similar women without HIV. While certain demographic characteristics are statistically different between groups, rates of substance use, an important predictor of health outcomes, are comparable. Experiences of social-structural disadvantages also differ but remain markedly high in both groups relative to the general population in Canada. For example, over a third of women in both groups reported a lifetime history of homelessness, while the national average is 2% for unsheltered and 15% for hidden homelessness [38]. This was achieved through targeted recruitment of our control group and is crucial to appropriately interpret any association with HIV.

## Limitations

These results must be considered in light of several limitations. Most notably, our analysis utilized cross-sectional clinical estimates of concurrent conditions, rather than confirmed medical diagnoses, to allow for inclusion of women receiving care outside of the study recruitment site and/or women not linked to care. While the clinical estimates chosen are objective and widely used in research, their use in isolation does not equate to comprehensive medical histories and clinical guidelines for diagnosis. As a result, the cross-sectional nature of our analysis is not optimal for the estimation of conditions that require serial laboratory tests/clinical visits for proper diagnosis, such as CKD and depression, or can change/improve over time (e.g., liver fibrosis). Next, our use of self-reported validated questionnaires for the clinical estimates of depression and PTSD may be limited by how questions are interpreted, recall bias of diagnoses, experiences of stigma

around mental health, and social desirability bias. These biases are likely to skew estimates toward underreporting, thereby attenuating observed agreement between self-reported and clinically derived measures rather than inflating concordance. The magnitude of this bias is expected to vary by condition, with greater impact for low-prevalence or asymptomatic conditions and those requiring longitudinal assessment. Importantly, 71% of women living with HIV in our study are connected to the Oak Tree Clinic, a specialized centre practicing comprehensive women-centred HIV care [39]. While observations in this analysis appeared to be condition-specific, rather than associated with HIV status, suggesting that the care site does not necessarily affect the findings, we acknowledge that our findings may not be generalizable to populations facing different socio-structural challenges than our cohort participants. An additional limitation is the absence of sensitivity analyses using alternative agreement indices or subgroup-specific analyses (e.g., by age or ethnicity), which may be informative in the presence of skewed prevalence and should be explored in future studies. Lastly, we were not able to include some conditions that are highly relevant for women with HIV, such as lung disease, osteoporosis, dyslipidemia, or cardiovascular disease. Given these limitations, future studies should validate self-reported and clinically derived measures against confirmed clinical diagnoses and adopt longitudinal designs to better characterize concordance over time. Where feasible, linkage to medical records and evaluation of health literacy interventions may help identify and reduce sources of discordance.

## Implications and future directions

Our findings of suboptimal agreement between self-reported data and clinical estimates for several concurrent conditions suggest that integrating both types of data may be a more effective way to accurately gauge the prevalence of physical and mental concurrent conditions in observational studies enrolling women living with HIV, when clinical charts are unavailable. Clinically, discordance may highlight opportunities to strengthen communication around diagnoses and screening results, particularly for mental health conditions and asymptomatic infections. From a public health perspective, these findings support targeted screening and knowledge-mobilization strategies that address stigma and structural barriers to disclosure. Enhancing the concordance between self-reported and clinical data not only increases the utility of self-report-based HIV research, it also empowers women, thereby contributing to more effective public health.

## Conclusions

In this cohort of women living with and without HIV in British Columbia, self-reported lifetime HCV history showed high concordance with serologic measures, whereas agreement for other physical and mental health conditions with clinical estimates was lower, highlighting opportunities to improve screening and communication for liver disease, kidney disease, depression, PTSD, and HBV. Such discordance may bias prevalence estimates in research, obscure unmet clinical needs, and complicate public health planning when self-reported data are used in isolation. The alignment between self-reported data and clinical estimates presents challenges, and while enhancing health literacy could empower women, the factors influencing these discrepancies are more complex than communication alone. Strategies to improve concordance include integrating self-reported, clinical, and serological data, adopting longitudinal study designs to assess concordance over time, and incorporating community-based knowledge mobilization to share findings with participating communities and improve health literacy for both personal wellbeing and research accuracy.

## Supporting information

**S1 Appendix. Demographic characteristics of women living with HIV and without HIV in the BCC3 study.**
(DOCX)

**S2 Appendix. Missing data in self-reported data and clinical estimates.**
(DOCX)

## Acknowledgments

We are thankful to all women who participated in the BCC3 study and made this work possible. This work was conducted on behalf of the BCC3 study team primarily on the traditional, ancestral, and unceded territories of the Coast Salish peoples, including the Sḵwx̱wú7mesh (Squamish), Səl̓ílwətaʔ/Selilwitulh (Tsleil-Waututh), and xwməθkwəy̓əm (Musqueam) Nations. In addition to the authors listed here, the BCC3 study team includes co-principal investigators Valerie Nicholson and Drs. Jason Brophy, Neora Pick, Allison Carter, Carmen Logie, Kate Salters, Mona Loutfy, Jerilynn Prior, and Joel Singer, as well as trainees/research staff Beheroze Sattha, Izabella Gadawska, Charity Mudhikwa, Julliet Kien Zama, Loulou Cai, Dr. Monika Kowatsch, Davi Pang, Melanie Lee, Marcela AP Silva, Shayda A Swann, Franceska Dnestrianschii, Amber R. Campbell, Vyshnavi Manohara, Sofia Levy, and Camille Valbuena.

## Author contributions

**Conceptualization:** Tovan Lew, Tetiana Povshedna, Hélène CF Côté.

**Formal analysis:** Tovan Lew.

**Funding acquisition:** Angela Kaida, Melanie CM Murray, Hélène CF Côté.

**Methodology:** Tovan Lew, Tetiana Povshedna.

**Supervision:** Hélène CF Côté.

**Visualization:** Tovan Lew.

**Writing – original draft:** Tovan Lew.

**Writing – review & editing:** Tovan Lew, Tetiana Povshedna, Elizabeth M King, Shelly Tognazzini, Angela Kaida, Melanie CM Murray, Hélène CF Côté.

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
