## [Decision Letter · Decision Letter 0]

23 Dec 2025

Thank you for submitting your manuscript to PLOS ONE. After careful consideration, we feel that it has merit but does not fully meet PLOS ONE’s publication criteria as it currently stands. Therefore, we invite you to submit a revised version of the manuscript that addresses the points raised during the review process.

Please submit your revised manuscript by Feb 06 2026 11:59PM. If you will need more time than this to complete your revisions, please reply to this message or contact the journal office at plosone@plos.org . . A letter that responds to each point raised by the academic editor and reviewer(s). You should upload this letter as a separate file labeled 'Response to Reviewers'.A marked-up copy of your manuscript that highlights changes made to the original version. You should upload this as a separate file labeled 'Revised Manuscript with Track Changes'.An unmarked version of your revised paper without tracked changes. You should upload this as a separate file labeled 'Manuscript'.

We look forward to receiving your revised manuscript.

Kind regards,

Ennio Polilli

Academic Editor

PLOS One

Journal Requirements:

“The British Columbia CARMA-CHIWOS Collaboration (BCC3) has received funding support from the Canadian Institutes of Health Research (CIHR) project grants (PJT-162348, PJT-175006), Community-based research grant (CBR-170103), and Women’s Health and Mentorship grant (F19-05017); CIHR HIV Clinical Trial Network support (CTN 335) and pilot grant funding (CTNPT046, CTNPT050); UBC Partner Recognition Fund; UBC Community University Engagement Support Fund, and UBC Public Scholar Initiative; Simon Fraser University’s Community Engagement Initiative; and Michael Smith Foundation of Health Research (trainee award). AK received a salary support award from the Canada Research Chair Program. MCMM received salary support from Michael Smith Foundation of Health Research Health Professional investigator award. TP received funding from the UBC Four Year Doctoral Fellowship, UBC Centre for Blood Research, and Mitacs Globalink Graduate Fellowship. TAGL received funding from the UBC Edwin S.H. Leong Centre for Healthy Aging Summer Student Research Award.”

4. One of the noted authors is a group or consortium: British Columbia CARMA-CHIWOS Collaboration (BCC3; CIHR CTN 335)

In addition to naming the author group, please list the individual authors and affiliations within this group in the acknowledgments section of your manuscript. Please also indicate clearly a lead author for this group along with a contact email address.

Reviewers' comments:

Reviewer's Responses to Questions

**Comments to the Author**

1. Is the manuscript technically sound, and do the data support the conclusions?

Reviewer #1: Yes

Reviewer #2: Yes

Reviewer #3: Partly

2. Has the statistical analysis been performed appropriately and rigorously?

Reviewer #1: Yes

Reviewer #2: Yes

Reviewer #3: Yes

3. Have the authors made all data underlying the findings in their manuscript fully available?

Reviewer #1: No

Reviewer #2: Yes

Reviewer #3: No

4. Is the manuscript presented in an intelligible fashion and written in standard English?

Reviewer #1: Yes

Reviewer #2: Yes

Reviewer #3: Yes

Reviewer #1: 1. Summary of the research

This manuscript investigates the concordance between self-reported and clinically assessed prevalence of selected chronic conditions and latent viral infections among women living with and without HIV in British Columbia, Canada. Using data from the BCC3 cohort, the study compares self-reported diagnoses of chronic kidney disease, liver disease, depression, post-traumatic stress disorder, hepatitis B, and hepatitis C with clinical estimates derived from screening tools and serological testing.

The authors report that agreement between self-report and clinical/serological measures varies by condition, with near-perfect concordance for HCV but only slight to fair agreement for other conditions. Cohen’s kappa statistics are used to quantify agreement. Demographic characteristics of the cohort are described.

The manuscript highlights challenges in relying solely on self-reported data for epidemiological research and clinical management, especially for conditions with low concordance. The authors recommend integrating self-report and objective measures to improve prevalence estimates and suggest that future research should address methodological limitations, explore subgroup differences, and provide more actionable recommendations for improving concordance.

Overall, the study addresses an important gap in understanding the reliability of self-reported health data in women living with and without HIV.

To strengthen the manuscript, the authors need to provide more comprehensive and transparent reporting of agreement metrics and statistical methods, expand the interpretation of findings by exploring causes of concordance and discordance, integrate demographic and contextual factors into the discussion, critically appraise study limitations, and offer concrete, actionable recommendations for future research and practice.

My overall recommendation is that the manuscript needs major revisions to meet PLOS ONE standards for transparency and methodological rigor.

2. Examples and evidence

2.1. Major technical issues

2.1.1. Title

a. The title is somewhat generic and not easy to read with the phrase "six commonly used clinical estimates or serological measures". The authors should consider specifying the population studied, what is being studied in short, and the country context

2.1.2. Abstract

a. Study Design Not Specified: The abstract does not indicate the study design. The authors will need to specify the study type in the methods.

b. Inconsistent Sample Sizes: There is a discrepancy in reported sample sizes between the abstract and the results section. The authors should clarify the final analyzed sample sizes, explain any exclusions or missing data, and ensure consistency throughout the manuscript.

c. Ambiguous Interpretation of Cohen’s Kappa: The abstract uses the phrase “fair to none” to describe agreement, which is non-standard and potentially confusing. It is recommended to report actual kappa values (with 95% confidence intervals if possible) for each condition, or at least the range. If qualitative descriptors are used (e.g., “fair,” “almost perfect”), the authors could cite a recognized interpretive scheme (such as Landis & Koch) and clarify these are heuristic. They should avoid using “none” as a category.

d. Statistical Reporting Lacks Precision: P-values and kappa values are presented without confidence intervals or details on the statistical approach. The authors should consider adding 95% confidence intervals for kappa values and clarify how agreement was assessed.

e. Unclear Definition of “Clinical Estimate”: The term “clinical estimate-based prevalence” is used without specifying that these are derived from screening tools or serology, not clinical diagnoses. The authors may need to briefly clarify what constitutes a “clinical estimate” in this context.

f. Abstract Conclusions Too General: The conclusions are vague (lines 62-63 and lines 64-65). The authors could be more specific, avoiding use of “may”, and indicating which conditions showed the greatest discordance and the practical implications for researchers and clinicians.

g. Missing Keywords: The listed keywords do not include some important methodological terms. To improve the manuscript’s discoverability and better reflect its content, it is suggested that the authors consider adding such terms, including “concordance,” “agreement,” and “Cohen’s kappa”.

2.1.3. Introduction

a. Clarity and Specificity of Study Rationale: The introduction provides a broad background on the health disparities faced by women living with HIV compared to men, and the importance of accurate prevalence estimates while the study assesses concordance of self-reports vs clinical estimates or serology among women living with HIV vs those without HIV. It does not clearly articulate the specific gap in the literature regarding the concordance between self-report and clinical/serological measures for the selected conditions among women living with HIV or without HIV, nor does it explain why this comparison is particularly important for the study population. I would therefore recommend that the authors revise the introduction to address these issues.

b. Justification for Use of Cohen’s Kappa: The introduction does not mention the methodological approach (i.e., use of Cohen’s

kappa or other concordance metrics) or why it is appropriate for the research question. It is suggested that the authors briefly introduce the concept of concordance/agreement and the rationale for using kappa statistics to assess the alignment between self-report and clinical/serological measures. They should also consider referencing methodological challenges (e.g., the “kappa paradox,” prevalence effects) and why robust agreement metrics are needed.

c. Insufficient Context from Prior Literature: While the introduction mentions general discrepancies between self-report and clinical data and that studies comparing self-report to clinical estimates are “scarce”. It however does not sufficiently reference prior studies that have specifically examined concordance using kappa or similar metrics, especially in HIV or similar populations. The authors should consider citing and briefly summarizing key findings from previous studies that have assessed agreement between self-report and clinical/serological data and highlight how this study builds on or differs from these prior studies.

d. Ambiguity in What Is Being Compared: The Introduction refers to “self-report-based prevalence” and “clinical estimates” but does not clarify how “clinical estimates” are derived vis a vis the screening tools or serology. The authors could clearly define in the introduction how “clinical estimates” are derived to the ensure readers can more easily interpret the results and limitations.

e. Objective Statement Needs More Precision: The objective is not very specific. The authors could make it make it more precise by specifying the conditions/viruses being compared, the population, and the methodological approach.

2.1.4. Methods

a. Study Design Not Indicated: As indicated in the abstract comments, the methods do not specify the study design, as required in the STROBE checklist. For transparency and context, authors should explicitly specify the study design in the title, abstract and methods of the manuscript.

b. The authors’ use of the term “outcome” in the abstract and methods for describing chronic conditions and infections measured at baseline is not ideal as it typically refers to results or endpoints following an intervention or exposure. The authors should consider using alternative terms to improve clarity and align with standard epidemiological terminology, ensuring readers understand that the study is assessing prevalence rather than outcomes of an intervention.

c. Incomplete Reporting of Operational Definitions and Cutoffs: The methods section refers to validated tools (KFRE, FIB-4, CES-D-10, PLC-C) but does not specify the exact cutoffs or criteria used to define a positive case for each condition. The authors could explicitly state the thresholds or scoring criteria for each tool with references for reproducibility and transparency.

d. Lack of Sample Size Justification or Power Consideration: There is no mention of how the sample size was determined, or whether the study is sufficiently powered to detect meaningful differences in kappa or prevalence. The authors may consider including a brief rationale for the sample size, even if post hoc, or discuss the precision of kappa estimates.

e. Issues with Kappa Statistic Reporting and Interpretation: The Methods state that “agreement was assessed using Cohen’s kappa” but do not specify whether unweighted or weighted kappa was used, nor the rationale for the choice. There is also no mention of reporting 95% confidence intervals for kappa, which is necessary for robust interpretation, and the interpretation framework for kappa (e.g., Landis & Koch) is not specified. The authors could clarify whether kappa was unweighted or weighted and justify the choice, and they should consider including 95% CIs for rigorous statistical reporting for reproducibility and appraisal. They should also consider pre-specifying the interpretive framework for kappa values (e.g., Landis & Koch) and justify its appropriateness for their context. Furthermore, the authors could consider reporting observed and expected agreement (p₀, pₑ) and prevalence/bias indices.

f. Ambiguity in Statistical Testing for Paired Data: The authors mention use of chi-squared and Fisher’s exact tests for categorical variables, but do not clarify what test was used for paired binary data (self-report vs. clinical estimate for the same individual). The authors could therefore specify the statistical test used for comparing paired proportions.

g. Software Version and Reproducibility: The methods mention GraphPad Prism 10.2.3, but do not specify the exact procedures or code used for kappa calculation and CIs. The authors could therefore briefly describe the procedure or provide a reference to the software’s method for kappa and CI calculation.

2.1.5. Results

a. Incomplete and Unclear Reporting of Agreement Metrics: The results section focuses on Cohen’s kappa and p-values but omits other important agreement metrics such as percent agreement, positive/negative agreement, and confidence intervals for kappa. These additional metrics are essential for interpreting the magnitude and reliability of agreement, especially when prevalence is skewed. To improve, the authors should consider also reporting percent agreement and, where appropriate, positive and negative agreement for each condition and group. They could also include 95% confidence intervals for all kappa statistics and briefly comment on the impact of prevalence and bias on kappa values to enhance interpretability.

b. Table 1: The table reports p-values and Cohen’s kappa values but does not provide confidence intervals for kappa, nor does it specify which statistical tests were used for each comparison. Qualitative descriptors for kappa are used without referencing a standard interpretive scheme. It is suggested that the authors revise the table to include 95% confidence intervals for all Cohen’s kappa values in the table, clearly indicate which statistical test was used for each p-value (e.g., Mann-Whitney U, chi-square, Fisher’s exact) and reference a recognized interpretive scheme for kappa descriptors (e.g., Landis & Koch) in the table legend or footnotes.

c. Readability and Transparency in Statistical Reporting: The results are densely presented, with minimal narrative synthesis and unclear statistical methods or handling of missing data. To improve clarity, the authors could summarize key findings in a concise narrative, specify the statistical tests used and any p-value adjustments, and clearly state how missing data were handled and their potential impact on the findings.

d. Lack of Sensitivity or Subgroup Analyses: No sensitivity or subgroup analyses are reported, which are recommended when prevalence is skewed, or agreement differs by subgroup. If feasible, the authors should consider reporting sensitivity analyses using alternative agreement indices and presenting agreement metrics stratified by key subgroups such as age or ethnicity. If the authors do not perform such analyses, they may need to acknowledge this as a limitation and suggest future research.

2.1.6. Discussion

a. Limited Analysis of Concordance and Discordance Across Conditions: The discussion summarizes the main findings but does not sufficiently analyze why certain conditions (e.g., HCV) show strong agreement between self-report and clinical/serological measures, while others (e.g., depression, PTSD, HBV) do not. To improve, the authors could expand the discussion to explore possible mechanisms for condition-specific differences in concordance, considering psychosocial, healthcare system, and other relevant factors. Comparing the magnitude of kappa values and prevalence discrepancies, and highlighting similarities and differences, would provide a more nuanced interpretation and better contextualize the findings.

b. Insufficient Integration of Demographic Characteristics, Generalizability, and Literature: Although demographic characteristics are reported in the results, they are not explicitly used in the discussion to assess generalizability or explore sources of bias or variation in concordance. Additionally, engagement with the broader literature on concordance between self-report and clinical/serological measures is limited. I would recommend that the authors discuss how demographic factors (e.g., age, ethnicity, socioeconomic status, substance use) may influence self-report accuracy and concordance, reflect on the applicability of the findings to other populations, and expand the literature review to include more studies on concordance, especially in similar populations.

c. Insufficient Integration with Broader Literature: The discussion references a few studies to contextualize findings but overall, it lacks a deeper engagement with the broader literature on concordance between self-report and clinical/serological measures. I would suggest that the authors expand the literature review to include more studies on self-report vs. clinical/serological concordance, especially in similar populations (e.g., women, people living with HIV, marginalized groups).

d. Superficial Treatment of Study Limitations and Methodological Constraints: The limitations section acknowledges some constraints but does not sufficiently address potential biases (e.g., selection bias, recall bias, social desirability bias) or discuss their impact on the interpretation of findings. The authors could provide a more nuanced discussion of potential biases, including how participant selection and self-reporting may affect generalizability and internal validity. They could also discuss the possible direction and magnitude of bias introduced by these limitations and suggest methodological improvements for future studies, such as longitudinal designs, validation against medical records, and inclusion of additional conditions.

e. Lack of Practical Implications and Specific Recommendations for Future Research and Practice: The discussion does not sufficiently address the practical implications of the observed concordance or discordance for epidemiological research, clinical management, or public health policy, particularly in marginalized populations. Recommendations for future research are somewhat generic and do not provide concrete suggestions for improving concordance or addressing identified gaps. The authors should elaborate on how their findings could inform future research design, clinical screening practices, and public health interventions, and offer specific recommendations for improving concordance and addressing the limitations identified.

2.1.7. Conclusion:

a. Overstatement of Findings: The conclusion overgeneralizes by stating that “women living with and without HIV in British Columbia are well-informed about their HCV status,” without adequately considering the study limitations. To authors should consider tempering the conclusion and explicitly acknowledging that concordance was lower for other conditions, and discussing the implications for research and clinical practice.

b. Insufficient Emphasis on Implications of Discordance: While the conclusion notes challenges in aligning self-reported and clinical data, it does not clearly articulate why discordance matters or its impact on research, clinical care, and public health. The authors should clarify the consequences of poor concordance and suggest how integrating self-report with clinical and serological data could enhance accuracy and utility in future studies and practice.

c. Lack of Specific Recommendations:

The conclusion calls for “community-based knowledge mobilization” and “improving health literacy,” but lacks concrete, actionable recommendations for researchers, clinicians, or policymakers. The authors should provide specific guidance, such as strategies for improving concordance, recommendations for future research design, or targeted interventions to address identified gaps.

2.2. General comments (relevant to all sections)

a. Data Availability Statement Not PLOS ONE Compliant: The Data Availability statement (“data available upon reasonable request”) does not meet PLOS ONE’s requirement for public sharing of the minimal dataset. It is recommended that the authors consider addressing this issue.

b. Manuscript Structure and Flow: The manuscript’s overall structure and flow would benefit from greater focus and logical progression across all sections. Specifically:

o In the introduction, transitions between the background, knowledge gap, and study objectives are unclear. The authors could revise the introduction for better logical flow and clear transitions for smoother progression.

o The Results section is dense and would be improved by using clearer subheadings or bullet points to highlight key findings by condition.

o The discussion section does not transition well with regards to ideas about interpretation of findings, limitations and implications, which could make it difficult for readers to follow the argument and understand the significance of the results. The authors could better organize the discussion into well-defined themes with clear transitions, ensuring that each idea logically follows from the previous one to enhance readability and clarity.

Addressing these points throughout the manuscript will help guide the reader, enhance clarity, and align the work with PLOS ONE guidelines for scientific writing.

c. Language, Clarity and Conciseness: Some sentences in the manuscript are long and complex, which can hinder understanding—for example, those found in lines 108–110 and 184–187. Additionally, there is frequent use of passive voice, as seen in lines 113–114 and line 148. To enhance clarity, the authors could break up lengthy sentences, use straightforward and direct language, and favor active voice. The authors could ensure that each sentence communicates a single, clear idea, and they remove redundant phrases to maintain conciseness.

d. Terminology Consistency: There is lack of consistency in the use of some terms in the manuscript. For example, the interchangeable use of the terms “concurrent conditions” and “comorbidities” throughout the manuscript may cause confusion. For greater consistency and clarity, the authors should select one term and use it uniformly. Additionally, qualitative descriptors for statistical measures like Cohen’s kappa should follow a recognized interpretive scheme and be applied consistently, ensuring readers can easily understand and compare results.

Reviewer #2: This article is well written and enphasizes the need for enhancing health literacy and improving communication between women and their care providers.

The limits of the study are accounted for. The conclusions are based ont the results.

I have minor comments that may be due to my misunderstanding, but I would not define as "fair" agreements with a Kappa score below 0.4 (for example lines 161-162), as in my knowledge it would be classified as "poor" or "weak".

Reviewer #3: The study sounds alright in the way it is presented. However the major downside is the unavailability of access to primary data used in the research. I would highly recommend that the researchers do the needful to have approval rights for the audience to access the primary data. If the primary data contains sensitive client details, kindly blind such details and give your readers confidence into what you have analysed.

**Do you want your identity to be public for this peer review?** For information about this choice, including consent withdrawal, please see our For information about this choice, including consent withdrawal, please see our Privacy Policy .

Reviewer #1: **Yes:** Brain C Chirombo, MBChB, MPHBrain C Chirombo, MBChB, MPH

Reviewer #2: **Yes:** CUZIN LiseCUZIN Lise

Reviewer #3: No

---

## [Author Response · Author response to Decision Letter 1]

4 Mar 2026

Manuscript reference number: PONE-D-25-43093

Title: Concordance between self-report and six commonly used clinical estimates or serological measures in health research

Dear Dr. Ennio Polilli and reviewers,

Thank you for your thoughtful review of our manuscript. We revised the manuscript to reflect the reviewer’s suggestions and provide a point-by-point response to the comments below. Revised parts of the manuscript are highlighted within quotations in blue.

Journal Requirements:

Thank you for bringing this to our attention. We have updated the manuscript in accordance with PLOS ONE guidelines.

“The British Columbia CARMA-CHIWOS Collaboration (BCC3) has received funding support from the Canadian Institutes of Health Research (CIHR) project grants (PJT-162348, PJT-175006), Community-based research grant (CBR-170103), and Women’s Health and Mentorship grant (F19-05017); CIHR HIV Clinical Trial Network support (CTN 335) and pilot grant funding (CTNPT046, CTNPT050); UBC Partner Recognition Fund; UBC Community University Engagement Support Fund, and UBC Public Scholar Initiative; Simon Fraser University’s Community Engagement Initiative; and Michael Smith Foundation of Health Research (trainee award). AK received a salary support award from the Canada Research Chair Program. MCMM received salary support from Michael Smith Foundation of Health Research Health Professional investigator award. TP received funding from the UBC Four Year Doctoral Fellowship, UBC Centre for Blood Research, and Mitacs Globalink Graduate Fellowship. TAGL received funding from the UBC Edwin S.H. Leong Centre for Healthy Aging Summer Student Research Award.”

Thank you for this comment. We have included this statement in our updated funding section.

Thank you for raising this important point regarding long-term data access. To ensure compliance with PLOS’ Data Availability Policy, we now provide a non-author institutional contact for data requests. Specifically, requests can be directed to the BCC3 email address (bcc3@cw.bc.ca), which is managed at the institutional level rather than by an individual author. The dataset will be securely stored on the institutional server, and a standard operating procedure has been established to ensure continuity of access regardless of changes in personnel. Our data availability statement is as follows:

Data availability statement:

The minimal dataset underlying the findings of this study cannot be made publicly available due to ethical and privacy restrictions related to participant confidentiality. A substantial proportion of the data is derived from individuals living with HIV, a highly stigmatized condition, and the dataset includes data from Indigenous participants, for whom principles of community consultation and governance apply prior to public data sharing. The data may be made available upon request through a non-author institutional point of contact at the BCC3 study office (bcc3@cw.bc.ca). Access will be granted in accordance with institutional ethics, privacy policies, and applicable Indigenous data governance principles. Procedures are in place to ensure continuity of access independent of the study authors.

4. One of the noted authors is a group or consortium: British Columbia CARMA-CHIWOS Collaboration (BCC3; CIHR CTN 335)

In addition to naming the author group, please list the individual authors and affiliations within this group in the acknowledgments section of your manuscript. Please also indicate clearly a lead author for this group along with a contact email address.

Thank you for this comment. The names of all individuals were listed in the acknowledgement section.

The acknowledgement on lines (337-343) reads:

“In addition to the authors listed here, the BCC3 study team includes co-principal investigators Valerie Nicholson and Drs. Jason Brophy, Neora Pick, Allison Carter, Carmen Logie, Kate Salters, Mona Loutfy, Jerilynn Prior, and Joel Singer, as well as trainees/research staff Beheroze Sattha, Izabella Gadawska, Charity Mudhikwa, Julliet Kien Zama, Loulou Cai, Dr. Monika Kowatsch, Davi Pang, Melanie Lee, Marcela AP Silva, Shayda A Swann, Franceska Dnestrianschii, Amber R. Campbell, Vyshnavi Manohara, Sofia Levy, and Camille Valbuena.”

Thank you for bringing this to our attention, the captions for Supporting Information files have been added to the end of the manuscript.

Lines (467-469):

“S1 Appendix. Demographic characteristics of women living with HIV and without HIV in the BCC3 study

S2 Appendix. Missing data in self-reported data and clinical estimates”

Reviewers' comments:

Reviewer's Responses to Questions

Comments to the Author

1. Is the manuscript technically sound, and do the data support the conclusions?

Reviewer #1: Yes

Reviewer #2: Yes

Reviewer #3: Partly

2. Has the statistical analysis been performed appropriately and rigorously?

Reviewer #1: Yes

Reviewer #2: Yes

Reviewer #3: Yes

3. Have the authors made all data underlying the findings in their manuscript fully available?

Reviewer #1: No

Reviewer #2: Yes

Reviewer #3: No

4. Is the manuscript presented in an intelligible fashion and written in standard English?

Reviewer #1: Yes

Reviewer #2: Yes

Reviewer #3: Yes

5. Review Comments to the Author

Reviewer #1: 1. Summary of the research

This manuscript investigates the concordance between self-reported and clinically assessed prevalence of selected chronic conditions and latent viral infections among women living with and without HIV in British Columbia, Canada. Using data from the BCC3 cohort, the study compares self-reported diagnoses of chronic kidney disease, liver disease, depression, post-traumatic stress disorder, hepatitis B, and hepatitis C with clinical estimates derived from screening tools and serological testing.

The authors report that agreement between self-report and clinical/serological measures varies by condition, with near-perfect concordance for HCV but only slight to fair agreement for other conditions. Cohen’s kappa statistics are used to quantify agreement. Demographic characteristics of the cohort are described.

The manuscript highlights challenges in relying solely on self-reported data for epidemiological research and clinical management, especially for conditions with low concordance. The authors recommend integrating self-report and objective measures to improve prevalence estimates and suggest that future research should address methodological limitations, explore subgroup differences, and provide more actionable recommendations for improving concordance.

Overall, the study addresses an important gap in understanding the reliability of self-reported health data in women living with and without HIV.

To strengthen the manuscript, the authors need to provide more comprehensive and transparent reporting of agreement metrics and statistical methods, expand the interpretation of findings by exploring causes of concordance and discordance, integrate demographic and contextual factors into the discussion, critically appraise study limitations, and offer concrete, actionable recommendations for future research and practice.

My overall recommendation is that the manuscript needs major revisions to meet PLOS ONE standards for transparency and methodological rigor.

2. Examples and evidence

2.1. Major technical issues

2.1.1. Title

a. The title is somewhat generic and not easy to read with the phrase "six commonly used clinical estimates or serological measures". The authors should consider specifying the population studied, what is being studied in short, and the country context

Thank you for this comment, we appreciate the need to make the title relevant and specific for readers. We have updated the title as follows:

“Concordance between self-report and six commonly used clinical estimates or serological measures: insights from a Canadian healthy aging study”

2.1.2. Abstract

a. Study Design Not Specified: The abstract does not indicate the study design. The authors will need to specify the study type in the methods.

Thank you for bringing this to our attention. We have revised the methods section to specify the study design. The abstract and Methods now clarify that women enrolled in the BCC3 Study, an interdisciplinary cohort study, were included in a cross-sectional analysis conducted between 2020 and 2024.

The abstract methods (lines 46-51) now reads:

“Women (aged ≥ 16 years residing in British Columbia) enrolled in the BCC3 Study, a prospective cohort, between 2020 and 2024 were included in the cross-sectional analysis. Self-reported prevalence of six conditions/viruses (chronic kidney disease, liver disease, depression, post-traumatic stress disorder, and hepatitis B and C viruses (HBV, HCV)), were compared to clinical estimates based on screening tools, and serology. Agreement was assessed via Cohen’s kappa.”

b. Inconsistent Sample Sizes: There is a discrepancy in reported sample sizes between the abstract and the results section. The authors should clarify the final analyzed sample sizes, explain any exclusions or missing data, and ensure consistency throughout the manuscript.

Thank you for this comment. We have revised the abstract to ensure consistency with the Results section by reporting the final analyzed sample sizes, rather than the total enrolled sample prior to application of exclusion criteria as was present before. Details regarding exclusions and missing data are now clearly described in the study flow diagram and Supplementary Materials.

Lines (53-54) now reads:

“In both women with (n=220) and without HIV (n=309), clinical estimate-based prevalence of depression and PTSD was higher than self-reported prevalence (all p<0.001).”

c. Ambiguous Interpretation of Cohen’s Kappa: The abstract uses the phrase “fair to none” to describe agreement, which is non-standard and potentially confusing. It is recommended to report actual kappa values (with 95% confidence intervals if possible) for each condition, or at least the range. If qualitative descriptors are used (e.g., “fair,” “almost perfect”), the authors could cite a recognized interpretive scheme (such as Landis & Koch) and clarify these are heuristic. They should avoid using “none” as a category.

Thank you for this comment. We have revised the abstract to remove the non-standard “none” and to align all qualitative agreement terminology with the Landis and Koch interpretive scheme, which is now cited in the Methods section. The descriptor has been replaced with “poor”, consistent with this framework, and agreement terminology is clarified as and interpretive scheme.

Lines (50-51 and 59-60) now reads:

“Agreement was assessed via Cohen’s kappa.”

“In both groups, agreement between self-report and clinical estimate of prevalence was fair to poor for all conditions/viruses (all κ<0.4), except for HCV, for which the agreement was near perfect (κ>0.8).”

d. Statistical Reporting Lacks Precision: P-values and kappa values are presented without confidence intervals or details on the statistical approach. The authors should consider adding 95% confidence intervals for kappa values and clarify how agreement was assessed.

Thank you for this comment. We have ensured that 95% confidence intervals for all Cohen’s kappa values are presented in Table 1, and the statistical approach used to assess agreement is described in detail in the Methods section, including the use of Cohen’s kappa to account for chance agreement and the interpretation framework applied. In the Abstract, we summarize agreement patterns using ranges rather than reporting individual kappa values and confidence intervals to maintain conciseness and readability, consistent with journal conventions, while directing readers to the full results table f

---

## [Decision Letter · Decision Letter 1]

19 Mar 2026

Concordance between self-report and six commonly used clinical estimates or serological measures: insights from a Canadian healthy aging study

PONE-D-25-43093R1

Dear Dr. Côté,

We’re pleased to inform you that your manuscript has been judged scientifically suitable for publication and will be formally accepted for publication once it meets all outstanding technical requirements.

Kind regards,

Ennio Polilli

Academic Editor

PLOS One

Additional Editor Comments (optional):

Reviewers' comments:

Reviewer's Responses to Questions

**Comments to the Author**

Reviewer #1: All comments have been addressed

Reviewer #3: All comments have been addressed

2. Is the manuscript technically sound, and do the data support the conclusions?

Reviewer #1: Yes

Reviewer #3: Yes

3. Has the statistical analysis been performed appropriately and rigorously?

Reviewer #1: Yes

Reviewer #3: Yes

4. Have the authors made all data underlying the findings in their manuscript fully available?

Reviewer #1: Yes

Reviewer #3: Yes

5. Is the manuscript presented in an intelligible fashion and written in standard English?

Reviewer #1: Yes

Reviewer #3: Yes

Reviewer #1: (No Response)

Reviewer #3: The responses sound worth considering, however in future you may need to make effort to get anonymised datasets to support your work.

**Do you want your identity to be public for this peer review?** For information about this choice, including consent withdrawal, please see our For information about this choice, including consent withdrawal, please see our Privacy Policy .

Reviewer #1: **Yes:** Brain C Chirombo, MBChB, MPH, FCOPHP-ECSABrain C Chirombo, MBChB, MPH, FCOPHP-ECSA

Reviewer #3: No

---

## [Editor Report · Acceptance letter]

PONE-D-25-43093R1

PLOS One

Dear Dr. Côté,

I'm pleased to inform you that your manuscript has been deemed suitable for publication in PLOS One. Congratulations! Your manuscript is now being handed over to our production team.

Kind regards,

on behalf of

Dr. Ennio Polilli

Academic Editor

PLOS One